# Deciphering Transcriptomic Variations in Hematopoietic Lineages: HSCs, EBs, and MKs

**DOI:** 10.3390/ijms251810073

**Published:** 2024-09-19

**Authors:** Swati Dahariya, Anton Enright, Santosh Kumar, Ravi Kumar Gutti

**Affiliations:** 1Department of Biochemistry, School of Life Sciences, University of Hyderabad, Hyderabad 500019, Telangana, India; swati@uohyd.ac.in (S.D.); skptl111@uohyd.ac.in (S.K.); 2Department of Pathology, University of Cambridge, Cambridge CB2 1QP, UK; aje39@cam.ac.uk

**Keywords:** hematopoiesis, hematopoietic lineages, transcriptome, lncRNAs

## Abstract

In the realm of hematopoiesis, hematopoietic stem cells (HSCs) serve as pivotal entities responsible for generating various blood cell types, initiating both the myeloid and lymphoid branches within the hematopoietic lineage. This intricate process is marked by genetic variations that underscore the crucial role of genes in regulating cellular functions and interactions. Recognizing the significance of genetic factors in this context, this article delves into a genetic perspective, aiming to unravel the biological factors that govern the transition from one cell’s fate to another within the hematopoietic system. To gain deeper insights into the genetic traits of three distinct blood cell types—HSCs, erythroblasts (EBs), and megakaryocytes (MKs)—we conducted a comprehensive transcriptomic analysis. Leveraging diverse hematopoietic cell datasets from healthy individuals, sourced from The BLUEPRINT consortium, our investigation targeted the identification of genetic variants responsible for changes in gene expression levels and epigenetic modifications across the entire human genome in each of these cell types. The total number of normalized expressed transcripts includes 14,233 novel trinity lncRNAs, 13,749 mRNAs, and 3092 lncRNAs. This scrutiny revealed a total of 31,074 transcripts, with a notable revelation that 14,233 of them were previously unidentified or novel lncRNAs, highlighting a substantial reservoir of genetic information yet to be explored. Examining their expression across distinct lineages further unveiled 2845 differentially expressed (DE) mRNAs and 354 DE long noncoding RNAs (lncRNAs) notably enriched among the three distinct blood cell types: HSCs, EBs, and MKs. Our investigation extended beyond mRNA to focus on the dynamic expression of lncRNAs, revealing a well-defined pattern that played a significant role in regulating differentiation and cell-fate specification. This coordination of lncRNA dynamics extended to aberrations in both mRNA and lncRNA transcriptomes within HSCs, EBs, and MKs. We specifically characterized lncRNAs with preferential expression in HSCs, as well as in various downstream differentiated lineage progenitors of EBs and MKs, providing a comprehensive perspective on lncRNAs in human hematopoietic cells. Notably, the expression of lncRNAs exhibited substantial cell-to-cell variation, a phenomenon discernible only through single-cell analysis. The comparative analysis undertaken in this study provides valuable insights into the distinctive genetic signatures guiding the differentiation of these crucial hematopoietic cell types.

## 1. Introduction

Hematopoiesis, the intricate process of generating various blood cell types, is crucial for maintaining the hematopoietic system’s functionality [1]. This process relies on complex gene expression patterns and regulatory networks. The coordination between protein-coding genes and noncoding RNAs plays a pivotal role in shaping the development and functions of different blood cell lineages [2]. This precise coordination is essential for maintaining hematopoietic balance and overall health. Hematopoietic stem cells (HSCs) possess distinct gene expression profiles that maintain their identity and regulate their differentiation into specific cell types [3]. Long noncoding RNAs (lncRNAs) are known to influence cell differentiation and development. However, our understanding of the complete lncRNA transcriptome in human hematopoietic cells is still limited. While a few lncRNAs have been identified to play roles in differentiation and development within the hematopoietic system, further research is needed to comprehensively define their functions and regulatory mechanisms.

Recent whole transcriptome sequencing has revealed a large number of putative lncRNAs. Notably, Cabezas-Wallscheid et al. [4] recently identified hundreds of lncRNAs expressed in HSCs. In addition, genomic profiling identified thousands of lncRNAs expressed in erythroid cells. lncRNA expression is tissue- and cell-type-specific [5,6,7,8] but is less conserved across species than in messenger RNA (mRNA) expression [9,10]. LncRNAs have been linked to the development of several lineages in hematopoiesis and in the immune response. Some lncRNAs were found to be enriched in HSCs [11] or dynamically expressed during erythropoiesis [12,13]. Some of them have been shown to play a role in erythroid maturation and erythro-MK development [12,13]. Despite these many examples [14,15,16,17,18,19,20] of specific functions for either stem cells or differentiated lineages such as myeloid and lymphoid cells, the repertoire of lncRNAs in human hematopoietic stem and progenitor cells has not been fully described.

Whole transcriptome sequencing allows large-scale profiling of lncRNAs in tissues and diseases, enabling the identification of many putative lncRNAs [8,21,22]. Recent advances in single-cell transcriptome profiling methods have unveiled unexpected variability in gene expression within seemingly homogeneous cell populations. Studies profiling lncRNAs at the single-cell level have revealed their cell-specific expression [8,23,24,25]. Previously, lncRNA expression was assessed by averaging transcriptomes of bulk RNA from mixed cell populations, limiting sensitivity to detect lncRNA expression in small cell populations and to resolve diversity within a cell type. Recent studies have disclosed lncRNA expression in purified murine MK-erythroid precursors, MKs, and EBs, as well as in human EBs, using deep sequencing. While differences in expression and function have not been extensively reported in HSCs, EBs, and MKs, they still remain largely unknown. Given that lncRNAs typically exhibit cell type or stage-specific expression and HSCs and MKs are scanty (≈0.01% of bone marrow) compared to EBs, many cell-type-specific lncRNAs may not have been identified and annotated yet. Therefore, our aim was to identify the full complement of lncRNAs expressed between HSCs, EBs, and MKs to determine lncRNAs specific to HSCs relative to representative differentiated lineages of EBs and MKs and to perform an initial analysis of their relevance to function.

In this study, we utilized datasets from the BLUEPRINT consortium, (https://projects.ensembl.org/blueprint/, accessed on 24 April 2019; fastq.gz format) covering the entire human genome across three blood cell types: HSCs, EBs, and MKs. Our primary objective was to analyze the expression profiles of both lncRNAs and mRNAs in these cell types, aiming to elucidate the potential roles of lncRNAs in HSCs, EBs, and MKs. Through de novo transcriptome reconstruction, we identified 3092 lncRNAs and discovered 14,233 potential novel trinity mRNAs previously unreported in public databases. Additionally, we characterized the distinct expression patterns of mRNAs and lncRNAs, constructing co-expression networks within HSCs, EBs, and MKs to uncover potential functional implications of mRNA and lncRNA expression in lineage-specific differentiation. Subsequently, Gene Ontology (GO) and Kyoto Encyclopedia of Genes and Genomes (KEGG) pathway analyses were performed to explore the functions of differentially expressed genes (DEGs) in HSCs, MKs, and EBs. Overall, our study provides a comprehensive assessment of lncRNA biology in the human hematopoietic system, suggesting their potential contributions to differentiation decisions during hematopoiesis and offering insights into identifying functional lncRNAs in other differentiation hierarchies.

## 2. Results

### 2.1. Analysis of RNA-Seq Data

To identify DE mRNAs and lncRNAs with expression profiles specific to each cell type in three different blood cell types: HSCs, MKs, and EBs; we have obtained Data Access Permission via a DAC agreement for the raw FASTQ mRNA-seq data. Following the removal of adaptor sequences, ambiguous nucleotides, duplicated sequences, poly-*N* reads containing >10% ‘N’, and low-quality bases (<Q30), a total of 1,449,823,626 cleaned reads (94.00%) were harvested for further analysis. The number of cleaned reads of each sample ranged from 13.4 to 116.2 million. The mapping ratio ranged from 88.0% to 94.0% with an average of 91.0%. A summary of read counts for all libraries and the percentage of reads mapped to the human genome for the biological replicates of HSCs, MKs, and EBs is tabulated (Appendix A).

### 2.2. Identification of DEGs (mRNAs and lncRNAs)

In order to capture the global gene expression profile associated with three different blood cell types (HSCs, MKs, and EBs), the BLUEPRINT RNA-seq data were utilized with a number of biological replicates, respectively. We employed the pipeline to detect and classify all mRNAs and lncRNAs within the expressed transcriptome [26]. By examining the expression profiles of three distinct hematopoietic lineages—we identified a total of 13,749 mRNAs and 3092 lncRNAs. Additionally, among these, 14,233 trinity mRNAs were identified (Appendix A). Following these initial examinations, we focused on the expressed transcriptome, covering 13,749 mRNAs and 3092 lncRNAs, to study gene expression across the three blood cell types: MKs, HSCs, and EBs. Using a log2 fold changes < −2 or >2 and a *p*-value cut-off of 0.01, we identified a total of 2845 DE mRNAs and 354 DE lncRNAs (Appendix A). We generated volcano plots (Figure 1) to provide an overview of the DEG patterns generated to represent upregulated and downregulated mRNAs/lncRNAs across different blood cell groups. Interestingly, we observed that in the mRNA data, there was a higher proportion of downregulated genes in all three blood cell lineages. Conversely, in the case of lncRNAs, the number of upregulated lncRNAs was higher compared to the number of downregulated lncRNAs across the same cell lineages (Appendix A). A Venn diagram was constructed to ascertain the overlap among the three DEG profiles of blood lineages (Figure 2).

Moreover, the analysis revealed 263 mRNAs exclusively DE in the HSCs/MKs comparison, 983 genes specific to the HSCs/EBs comparison, and 202 mRNAs specific to the MKs/EBs comparison (Figure 2a). Additionally, the study identified a total of 354 (159 + 48 + 18 + 77 + 23 + 22 + 7) DE lncRNAs (Figure 2b) across the three groups (MKs, HSCs, and EBs) (Using a log2 fold changes < −2 or >2 and a *p*-value cut-off of 0.01). Among them, 48 lncRNAs were specifically DE in the HSCs/MKs comparison, 159 lncRNAs were specific to the HSCs/EBs comparison, and 18 lncRNAs were specific to the MKs/EBs comparison (Appendix A) (Figure 2b). These specific DEGs provide valuable insights into the unique gene expression patterns characterizing each blood cell lineage. Venn diagrams provide a visual depiction of both the common and distinct DEGs among HSCs, MKs, and EBs. This analysis, driven by differential expression, revealed 137 shared mRNAs (Figure 2a) and 7 shared lncRNAs (Figure 2b) in the group of HSCs, EBs, and MKs. Notably, the discovery of 137 mRNAs/genes and 7 lncRNAs exhibiting differential expression and shared across all three cell lineages (MKs, HSCs, and EBs) points towards the existence of a shared regulatory network governing crucial biological processes in blood cell types.

This finding offers valuable insights into the unique gene expression alterations within each blood cell type. Notably, the majority of mRNAs in HSCs exhibited downregulation, indicating lower expression levels compared to other cell types or conditions (Appendix A). On the one hand, the DE mRNAs that were upregulated in HSCs displayed downregulation in EBs and MKs, indicating higher expression levels in HSCs but reduced levels in the other two cell types. Conversely, the DE lncRNAs, such as UCA1, AC100835.2, AC104561.3, LINC01764, and LINC00656, consistently showed a downregulation pattern across all three cell lineages (MKs, HSCs, and EBs), with the exception of LINC02573 and AL365361.1 lncRNAs, etc. These two lncRNAs exhibited a distinct expression pattern compared to the rest. To visually explore the expression patterns of both unique and common DE mRNAs and DE lncRNAs, unsupervised hierarchical clustering was employed to generate heatmaps (Figure 3 (Appendix A)). Furthermore, the heatmaps display a cluster of the top 137 mRANs and 7 lncRNAs that were DE among all blood cell samples. The analysis revealed that MKs and EBs showed greater similarity in terms of the genes DE, while the pattern of downregulation of lncRNAs was more pronounced in HSCs and MKs compared to EBs.

The comparative analysis of blood cells revealed that among the total, 2845 mRNAs and 354 lncRNAs were differentially expressed between all blood cell lineages. Within this set, 917 mRNAs and 276 lncRNAs were upregulated, while 2177 mRNAs and 102 lncRNAs were downregulated (Using a |FC| < −2 or >2 and a *p*-value cut-off of 0.01). Venn diagrams were plotted to depict the cumulative count of DE mRNAs and DE lncRNAs (Figure 4), including upregulated and downregulated instances. The diagram captures variances across various cell types and within each specific cell type within the blood.

Additionally, the analysis using the Venn diagram highlights the distribution of DE mRNAs and DE lncRNAs across distinct blood cell samples, offering insights into the prevalence of upregulated and downregulated mRNAs/lncRNAs. Between HSCs and MKs, a total of 1266 mRNAs showed differential expression, with 326 genes upregulated and 940 mRNAs downregulated. In the comparison between HSCs and EBs, there were 2186 DE mRNAs, including 500 upregulated mRNAs and 1686 downregulated mRNAs. The MKs vs EBs comparison resulted in 927 DE mRNAs, comprising 364 upregulated mRNAs and 563 downregulated mRNAs (Appendix A).

The highest number of DE mRNAs (2186) was observed between HSCs and EBs, with 500 genes upregulated and 1686 mRNAs downregulated. The differential expression between HSCs and MKs (1266 mRNAs) was similar to that between MKs and EBs (927 mRNAs), with 326/364 mRNAs upregulated and 940/563 mRNAs downregulated. However, the number of DE mRNAs increased when comparing EBs with other lineages, as compared to the comparison between MKs and EBs. Regarding DE lncRNAs, there were 154 common DE lncRNAs between HSCs and MKs, with 85 upregulated and 69 downregulated. For the HSCs vs EBs comparison, out of 266 DE lncRNAs, 213 were upregulated and 53 were downregulated. Between MKs and EBs, 48 DE lncRNAs were upregulated and 22 were downregulated, out of a total of 70 DE lncRNAs (Appendix A).

It’s worth noting that despite the presence of several thousand lncRNAs showing differential expression within each cell lineage; only 7 lncRNAs were common among all three blood cell types. The fewest common DE lncRNAs were observed between MKs and EBs, followed by HSCs and MKs, and HSCs and EBs. Therefore, the differential expression of mRNAs/lncRNAs was more pronounced in the comparisons between HSCs vs EBs and HSCs vs MKs than in MKs vs EBs.

### 2.3. DEGs Functional Enrichment Analysis

We performed functional enrichment analyses of GO and KEGG pathways using the bioinformatics resource DAVID. The goal was to gain insights into the biological processes and molecular functions influenced by the DEGs under investigation, as well as their cellular localization are presented in Figure 5a (Appendix A). The DEGs were classified into three primary categories: molecular function, biological process, and cellular component. Specifically, 177 DEGs were assigned to biological process terms, 143 DEGs to molecular function terms, and 153 DEGs to cellular component terms.

In the biological process category, the DEGs were categorized into various functional subcategories. Among these, the most abundant GO terms were associated with essential cellular processes, including metabolic processes, cellular component organization and biogenesis, regulation of biological quality, cell cycle processes, and response to external stimuli. Particularly interesting were the upregulated genes that encoded cyclins, which are key players in regulating the progression of the cell cycle, particularly during the G1-to-S and G2/M phase transitions. These genes likely play crucial roles in maintaining cellular homeostasis in response to chemical signals and external stimuli during cell division.

Regarding the molecular function term, the majority of the identified unigenes exhibited functions related to protein binding and carbohydrate derivative binding. Several genes within this category were both upregulated and downregulated, suggesting their involvement in essential cellular processes. Some of these genes were associated with receptor binding, anion binding, enzyme binding, small molecule binding, purine ribonucleoside triphosphate binding, enzyme regulator activity, and protein dimerization activity, highlighting their significance in cellular functions and interactions.

In the cellular component category, the DEGs were found to be associated with specific cellular locations, including the cytoplasm, cytoplasmic part, membrane, intracellular organelle lumen, and membrane-enclosed lumen. These findings provide insights into the cellular compartments where the genes of interest are predominantly active.

In Figure 5b, which illustrates the KEGG-pathway-enrichment analysis, we identified matches for 44 unigenes associated with 17 KEGG pathways. Our KEGG-pathway analysis unveiled 17 pathways enriched with our identified partner mRNAs, which were likely influenced by the activity of the 7 DE lncRNAs. Overall, the functional enrichment analysis revealed essential biological processes, molecular functions, and cellular localizations associated with the DEGs. The identification of key pathways involved in hematopoietic cell lineage.

### 2.4. Correlation between Coding Genes and lncRNAs

We examined the impact of lncRNAs on mRNA expression and their associated biological functions across three distinct cell lineages: HSCs, MKs, and EBs. To uncover significant interactions, we constructed correlation matrices between DE lncRNAs and DE mRNAs. These matrices are visualized as heatmaps for individual cells. We focused on interactions with a Pearson correlation coefficient (|r|) exceeding 0.9 and a *p*-value less than 0.05 (depicted in Figure 6 (Appendix A)). From these analyses, we identified 225 lncRNA–mRNA interactions with notable binding potential. Employing gene ontology and KEGG-pathway assessments, we focused on DE mRNAs and DE lncRNAs displaying significant correlations. Specifically, we investigated 225 target genes derived from the seven most prominent and recurrently observed lncRNAs to gain insights into their biological implications.

To visually illustrate the lncRNA–mRNA interactions and their functions, we constructed a co-expression alluvial plot (Figure 6a). This plot provided a clear representation of the predominant interactions between the identified lncRNAs and their corresponding mRNA targets, based on the frequency of interactions with each mRNA partner. Ninety-one target mRNAs have been prominently correlated with lncRNAs and ranked and are listed in detail (Appendix A). These mRNAs play essential roles in a wide array of molecular functions, including binding and activity functions. This diversity encompasses various types of bindings, such as protein, anion, carbohydrate-derivative, enzyme, and receptor bindings, among others. Additionally, activities such as protein dimerization, enzyme regulation, cytokine response, peptide interactions, and amide functions are also prevalent among these identified targets. Furthermore, the mRNAs that exhibit frequent correlations are associated with key biological processes, including anatomical structure development, animal organ development, biological regulation, cell communication, cell surface receptor signaling pathways, cellular processes, cellular response to stimuli, developmental processes, and more. Likewise, these mRNAs are implicated in various essential cellular processes, including the cell periphery, cytoplasmic vesicles, cytoplasm and its components, endomembrane systems, extracellular regions and parts, cytosol, intracellular non-membrane-bounded organelles, membrane-bounded organelles, membrane-enclosed lumens, as well as organelle parts and lumens, among other crucial cellular contexts.

### 2.5. Analysis of a Co-Expression Correlation Modules Using WGCNA

We employed the “WGCNA (v1.72-1)” R package (v4.3.1) to associate DEGs exhibiting comparable expression patterns with modules using the method of average linkage clustering. Utilizing the principles regarding cluster dendrograms, the analysis of the dendrogram reveals distinctive gene clusters organized based on their shared expression patterns, enabling the identification of co-regulated gene groups. In this specific investigation, the dendrogram resulting from cluster analysis, computed through correlation coefficients distance (depicted in Figure 7 panels), encompasses the dataset of samples from three distinct blood cell lineages. The analysis distinctly segregates genes into three primary clusters, employing a complete linkage method with a correlation coefficients distance, setting the similarity threshold at 70%. Among the co-expressed genes, three distinct clusters are discerned: the first cluster (brown), the second cluster (green), and the third cluster (blue).

Notably, the second and third clusters exhibit significantly heightened interrelationships among the samples derived from the three cell lineages, in contrast to the first cluster. The first cluster (brown) embodies a similarity level exceeding 80%, consolidating samples that are closely grouped alongside genes displaying a similarity level of approximately 70%. This grouping is founded on the akin characteristics observed in terms of the correlation coefficients distance of their co-expression parameters. “Akin characteristics” refers to similarities or resemblances between different elements. In the context of the sentence, it means that the genes in the first cluster are grouped together because they share similar or comparable characteristics in terms of their correlation coefficients distance of co-expression parameters. These genes likely exhibit common patterns of expression or behavior, which is why they are clustered together based on their shared traits. Cluster 2 (green) encompasses samples that are grouped together in a cohesive manner, representing an extraordinary similarity level surpassing 95%.

Similarly, Cluster 3 (blue) includes samples that are clustered in a concerted manner, indicative of a similarity level surpassing 95%. The amalgamation of samples in these clusters underscores their shared co-expression traits and highlights the coherent patterns present within their expression profiles. As shown in Figure 7a, by sample clustering, no outliers were observed in the respective samples, thus all samples were included in the further analysis.

In this study, we undertook comprehensive co-expression analyses using samples derived from three distinct blood lineages, each complemented by its corresponding clinical data. To precisely establish the co-expression relationships, we navigated through a process that hinged upon both scale independence and mean connectivity analyses. This exploration was conducted across modules, utilizing a spectrum of power values spanning from 1 to 20. The crux of this analysis lies in the astute selection of a soft-threshold power value, β = 10, which yielded a crucial outcome—a scale-free R^2^ value of 0.00. This R^2^ value is pivotal as it mirrors the underlying structure of our analysis, and this significant interplay is graphically depicted in Figure 7b. Our investigations ventured into understanding the interplay between scale independence and mean connectivity—a representation of gene connections within the co-expression network. Within this framework, a distinct set of patterns unfurled. Specifically, as we set the power value at 1 and 2, the corresponding scale independence values attained −0.01. Concurrently, the mean connectivity value rested at 1. This pattern signifies the effectiveness of these threshold values in approximating the desired scale-free structure.

Notably, this observation accentuates certain genes’ or gene groups’ remarkable interconnectedness despite a relatively low threshold power. This phenomenon alludes to the formation of dense interconnected clusters within the network. At threshold values of 3 and 4, the scale-free topology values manifested within the range of −0.05, coinciding with a mean connectivity value of 2. This alignment with scale-free distribution illustrates the potency of these thresholds in structuring the network differently. The mean connectivity value of 2 corresponds to a moderate level of connectivity, maintaining a balance between threshold power and average connections. However, for other threshold values, the scenario shifts. The absence of consistent or desired scale-free topology, within the range of −0.08 to 0.00, confines the mean connectivity values below 6. This observation holds significance, signifying that, as the co-expression threshold tightens, numerous genes fall short of meeting the criteria for robust connections. This intriguing trend underscores the intricate relationship between threshold power, connectivity, and network structure. In essence, our systematic investigation highlights the nuanced interplay between soft-threshold power, mean connectivity, and the formation of meaningful co-expression relationships among genes within diverse blood lineages. This exploration enriches our understanding of the intricate dynamics governing gene interactions and regulatory mechanisms within distinct biological contexts.

Upon computation, it is our assertion that for a correlation coefficient of 0.00 (with a soft threshold β of 10), the co-expression network exhibits a heightened correlation, rendering it more apt for the creation of distinct gene modules (depicted in Figure 7b). Utilizing both the TOM and the hierarchical average linkage clustering method, gene modules within each gene network were identified (using parameters deep split = 2, cut height = 0.4). The ensuing heatmap, presented in Figure 7c, visually represents these findings. Furthermore, the gene cluster tree was computed and the outcomes are showcased in Figure 7d. The network heatmap displays the relationships between different gene modules. Each row and column in the heatmap correspond to different gene modules. The shading within each cell reflects the degree of correlation or co-expression potency between the modules. Deeper hues denote heightened correlations, signifying a more frequent tendency for genes from those modules to be co-expressed jointly. Conversely, lighter shades or unoccupied cells imply diminished co-expression significance. This pattern becomes apparent in the MEbrown module, where genes inclined towards co-expression manifest as more robust connections. Meanwhile, genes associated with the MEturquoise and MEblue colors exhibit less pronounced or absent co-expression, resulting in weaker interconnections.

To conclude, the eigengene heatmap and network heatmap plots within the framework of WGCNA yield invaluable insights into the foundational gene expression patterns and inter-module connections. The eigengene heatmap elucidates how gene modules react under varying conditions, while the network heatmap exposes the interwoven nature of distinct modules, casting illumination upon plausible regulatory associations and biological mechanisms.

## 3. Discussion

To the best of our knowledge, this report represents the first catalog of the repertoire of protein coding and lncRNA elements underlying three blood cell lineages (HSCs, MKs, and EBs) in humans.

A recent evolutionary study revealed differences in the patterns of mouse lncRNAs between HSCs and differentiated lineages such as B cells and granulocytes. Recent whole transcriptome sequencing has unveiled a plethora of putative lncRNAs. The function of some lncRNAs has been established in a limited scope of biological processes, such as cell cycle regulation, embryonic stem cell pluripotency, lineage commitment and differentiation, and cancer progression [27,28,29,30]. In the hematopoietic system, only a few lncRNAs have been identified to be involved in the differentiation or development of hematopoietic lineages [31]. The roles of numerous lncRNAs have been explored in vitro models of hematopoietic multi-lineage differentiation, including granulocyte differentiation [17,32], eosinophil differentiation [15], and erythropoiesis [14].

Considering that lncRNAs usually exhibit cell type or stage-specific expression and (HSCs, MKs, and EBs) are rare (0.05% of bone marrow), we reasoned that many HSC-specific lncRNAs may not have been identified and annotated yet in humans.

Notably, Cabezas-Wallscheid et al. [4] recently identified hundreds of lncRNAs expressed in HSCs and compared their expression with that in lineage-primed progenitors. However, the current GENCODE annotation for lncRNAs predominantly relies on easily cultured cell lines or whole organisms, lacking many cell-type-specific hematopoietic transcripts. To address this limitation, certain research groups have compiled annotations for subsets of the hematopoietic lineage or specific differentiation models [11,12,13]. In a recent endeavor, a robust annotation was developed, encompassing cell types ranging from HSCs to differentiated cells, spanning both myeloid and lymphoid lineages, as well as blood cancers [33].

This study conducted an in-depth analysis of RNA-Seq data to identify DE mRNAs and lncRNAs in three distinct blood cell types: HSCs, MKs, and EBs. The goal was to unravel unique expression profiles specific to each cell type and gain insights into the regulatory networks governing their functions. Studying HSCs, MKs, and EBs presents numerous challenges due to their extremely low abundance and the absence of favorable culture conditions for their proliferation. In addition to this, BLUEPRINT was a cornerstone consortium. This consortium has devised protocols tailored for analyzing single or extremely small quantities of cells, facilitating the examination of rare to ultra-rare cell types, including clinically significant materials like biopsies. Consequently, BLUEPRINT has amassed approximately 3500 datasets encompassing healthy and diseased cell types [34].

Raw data from the BLUEPRINT epigenome project for HSCs, EBs, and MKs as healthy controls were accessed for this study. Through an in-depth RNA-Seq data analysis, we found 13,749 mRNAs and 3092 lncRNAs. Deep RNA-Seq data analysis followed by de novo transcriptome reconstruction was adopted for genome-wide annotation and functional characterization of novel lncRNAs. Among these, 14,233 trinity mRNAs, 120 were found, contributing to a comprehensive total of 31,074 transcripts in the expressed 121 transcriptome. We have uncovered several hundred newly discovered lncRNAs, with a significant portion showing high expression levels in the respective cell lineage, particularly in MKs. However, this combination of differential expression, synteny, and conserved expression is broadly applicable to other cell types in the hematopoietic lineage or other tissues. This list contains a number of lncRNAs not reported in a previous study but is more comprehensive. Furthermore, we performed a series of bioinformatics analyses to define those lncRNAs, including analysis of global lncRNA expression profiles, examining their conservation, overlap with repeats, and high correlation or anti-correlation with lncRNA expression with gene expression and marks in the current study.

The present results provide a comprehensive overview of the findings from the RNA-Seq data analysis conducted on the BLUEPRINT Consortium dataset. It effectively communicates the DE patterns observed in both mRNAs and lncRNAs across various blood cell types. The comparison between HSCs, MKs, and EBs sheds light on the dynamic regulation of gene expression during hematopoiesis. Additionally, highlighting the commonalities and exceptions within the expression patterns of both mRNAs and lncRNAs enhances the depth of understanding gained from the analysis. With a focus on lncRNA-mediated phenotype regulation, we examined RNA-Seq data from the BLUEPRINT Consortium. The analysis identified 1448 DE mRNAs (983 in HSCs, 263 in MKs, and 202 in EBs) and 225 DE lncRNAs (159 in HSCs, 48 in MKs, and 18 in EBs) across HSCs, MKs, and EBs. Within these cell types, we found 137 common mRNAs and 7 common lncRNAs. The majority of common mRNAs displayed upregulated expression, especially during differentiation from HSCs to EBs or MKs. However, the comparison between MKs and EBs revealed an irregular pattern of differential mRNA expression. Among the 7 common DE lncRNAs, downregulation predominated during the transition from HSCs to EBs or MKs. Notably; exceptions were observed with two lncRNAs, LINC02573 and AL365361.1, displaying irregular expression patterns across all cell types. We further investigated the expression correlation between 7 lncRNAs and their putative 137 mRNA targets, recognizing the regulatory role lncRNAs play in modulating mRNA expression [35].

Functional validation presents a formidable obstacle owing to the abundance of both annotated and novel lncRNAs. To explore the biological roles of these lncRNAs and their associated mRNAs, we have constructed the lncRNA–mRNA co-expression network and then conducted GO term and pathway-enrichment analyses for the highly co-related mRNAs. It is long acknowledged that the RNA-seq-based lncRNA–mRNA network has become a useful tool to predict functional lncRNAs and their potential functional mechanisms [36]. Of these interactions, most mRNAs are associated with distinct lncRNAs such as LINC02573 [37], AL365361.1 [38], LINC00656 [39], AC104561.3 [40], AC100835.2, LINC01764 [41], and UCA1 [42] (Appendix A). The upregulated mRNAs were significantly enriched in the enzyme binding process, carbohydrate derivative binding, purine ribonucleoside triphosphate binding, protein binding, enzyme regulator activity, developmental and biological regulation including cell-cycle cell communication, and response to various stimuli like stress, oxygen-containing compounds, etc. indicating that a variety of development-related genes were aberrantly upregulated and participated in the cellular biogenesis/organization process. The KEGG-pathway analysis unveiled that several of our DE lncRNAs and their mRNAs partners are prominently engaged in pathways related to cytokine–cytokine receptor interaction and cell adhesion molecules as well as cell cycle progression through p53 mediated [19] and NF-kappa B signaling pathways [43] which indicates that lncRNAs were also involved in the crucial regulating mechanism of hematopoietic cell lineage differentiation and development.

Certainly, the overarching challenges in lncRNA biology persist, particularly concerning the precise mechanisms by which noncoding RNAs regulate the differentiation, developmental potential, and metabolic capacity of hematopoiesis. Leveraging bioinformatics analysis of the transcriptome, we offer a comprehensive database poised to underpin future investigations into lncRNA biology within the human hematopoietic system. Together with the lncRNA–mRNA co-expression network, we believe that a comprehensive understanding of the complex networks of interactions that these DE RNAs coordinate would provide a unique opportunity for better therapeutic interventions.

## 4. Materials and Methods

### 4.1. Datasets Gathering

Unveiling the intricate process of hematopoietic cell differentiation involves observing gene expression changes as cells progress through maturation within the hematopoietic system’s hierarchical structure. To explore this, we accessed expression profiles from three distinct blood cell types—HSCs, MKs, and EBs—via the publicly available BLUEPRINT consortium (https://projects.ensembl.org/blueprint/). On 24 April 2019, we secured data access permission for the raw FASTQ datasets pertaining to MKs, HSCs, and EBs under the BluePrint Epigenome project. Our objective was to align these datasets with the latest annotations of lncRNAs and protein-coding sets. In the present study, a total of eight blood cell samples, including three MKs, two HSCs, and three EBs, spanning various ages and genders (male/female) were utilized for mRNA and lncRNA profiling. Appendix A provides the DataSet ID, study ID, sample accession, and raw file accession ID. Following stringent quality control measures, read mapping, and normalization of RNA-seq data, we computed the FPKM values for each gene and lncRNA. Additionally, we employed the Trinity algorithm for de novo discovery of lncRNAs, aiming to predict novel transcripts within the dataset.

### 4.2. Data Pre-Processing

In the pre-processing of raw sequencing data, we performed a quality assessment using FastQC. To improve read quality, we applied Trimmomatic (v.0.35) for adapter removal and quality trimming based on the FastQC results. The resulting clean reads met quality conditions (Q20 > 90% and Q30 > 85%). For alignment, we used Hisat2 with the human reference genome GRCh37 (hg19). BAM files representing mapped reads were generated using SAM tools (v.1.3.1), and unmapped reads were discarded. HTSeq (v.0.6.1p1) was then used with a GTF file from Ensembl (v.82) to count the number of reads aligning to each gene. HTSeq assigns reads to specific genes based on alignment positions, generating a count matrix with genes as rows and samples as columns. Subsequently, differential expression analysis was performed using DESeq2, which involved normalizing the counts and accounting for library size differences and technical biases. The gene expression values were represented using normalization techniques such as FPKM. Cuffdiff (v2.1.1) was used to calculate the FPKM of both the lncRNAs and coding genes in each sample [44,45,46].

### 4.3. Gene Expression Quantification and Differential Expression Analysis

In this study, we aimed to identify DEGs, including both mRNAs and lncRNAs, among different blood cell types: HSCs, MKs, and EBs. To achieve this, we utilized the BLUEPRINT gene expression dataset, which contains gene expression profiles of various hematopoietic cell lineages and is publicly available from the BLUEPRINT consortium website (https://projects.ensembl.org/blueprint/, accessed on 24 April 2019). These values were calculated using the R/Bioconductor (v 4.3.1) and Limma package (v3.26.8) from the GEO2R tool. The gene expression data sets were analyzed using the R/Bioconductor, specifically the Limma package (v3.26.8) from the GEO2R tool (https://bioconductor.org/packages/release/bioc/html/limma.html, accessed on 28 July 2024). This package provided the necessary tools for performing differential expression analysis using the *t*-test. The FDR of 5%, was calculated using the Benjamini and Hochberg method to control for multiple testing [47,48]. The DEGs were identified based on a log2FC > 2 and a *p*-value cut-off of <0.01. A volcano plot is a scatterplot that helps identify significant changes in gene expression between two groups or conditions (https://www.bioinformatics.com.cn/plot_basic_3_color_volcano_plot_086_en). It plots the log fold change (X-axis) against the statistical significance (Y-axis), usually represented as a *p*-value. This plot helps identify genes that exhibit large fold changes and significant statistical differences. Bioinformatics and Evolutionary Genomics online software (https://bioinformatics.psb.ugent.be/webtools/Venn/, accessed on 28 July 2023) was used to obtain the intersection of HSCs/MKs, HSCs/EBs, and MKs/EBs, and determine the gene expression by analysis between the intersection and union. In addition to the Venn and volcano plot, we also utilized other visualization techniques such as a heatmap to visualize the significant DEGs and lncRNAs shared among the three transgenic and distinct groups corresponding to HSCs, MKs, and EBs. To visually represent the significant DEGs, a heatmap was generated for each dataset using R. The heatmap is a visual representation that displays the expression patterns of the identified DEGs using a color-coding system.

### 4.4. Functional Enrichment Analysis

In the functional enrichment analysis of DEGs, GO and KEGG were used to identify biological processes, molecular functions, cellular components (for GO), or metabolic and signaling pathways (for KEGG) that are statistically over-represented in the set of DEGs. The principle behind this analysis is to compare the identified DEGs in order to detect categories that are significantly enriched. By identifying these over-represented categories, we can infer that certain biological processes or pathways are either activated or suppressed in the specific experimental condition being studied. The objectives of GO and KEGG enrichment analyses are to do the following: (i) characterize the biological roles of the DEGs by mapping them to established functional categories and (ii) identify key biological processes and pathways that are significantly enriched in the experimental condition, helping to reveal potential biological mechanisms related to disease progression, treatment response, or other phenomena. By understanding the biological context of the DEGs, these analyses provide insights into the functional implications of gene expression changes under various conditions.

The GO enrichment analysis, accessible at http://geneontology.org/page/go-enrichment-analysis, and DAVID annotation from https://david.ncifcrf.gov/, accessed on 28 July 2023 for functional annotation and pathway analysis. This allowed them to investigate the MFs, BPs, and CCs associated with the genes of interest. They considered GO terms to be significantly enriched within the gene set if they had an FDR of less than 0.05. Moreover, the distinctively regulated genes were organized into gene pathways through pathway enrichment analysis utilizing the KEGG. This approach aimed to shed light on the potential pathways influenced by the genes under investigation. The DEGs found to be statistically enriched in KEGG pathways were considered significant with a corrected *p*-value of less than 0.05.

### 4.5. Correlation Analysis

Correlation analysis was conducted to examine the relationship between lncRNAs and mRNAs in blood cell samples. DE lncRNAs were utilized to predict potential target genes. Normalized counts of these lncRNAs and mRNAs were subjected to Pearson correlation analysis, yielding the PCC. By calculating the PCC, co-expressed lncRNAs and mRNAs were identified. Significant lncRNA–mRNA interactions were determined using a threshold of Pearson’s correlation coefficient |r| ≥ 0.94 and *p*-values of at least 0.001. The resulting correlations were visualized in a heatmap using R programming.

### 4.6. Co-Expression Network Construction by WGCNA Analysis

A co-expression network for all genes in blood cell samples was constructed using the “WGCNA (v1.72-1)” R package (v4.3.1). The algorithm filtered out genes with the top 25% variance for further analysis. One WGCNA analysis was performed on the samples, where Pearson’s correlation matrices were calculated using Equation (1).
amn = |cmn|β(1)
where amn represents the adjacency between gene m and gene *n*, cmn denotes Pearson’s correlation, and β represents the soft-power threshold), a weighted adjacency matrix was created. The power parameter ranging from 1–20 was screened out using the pick SoftThreshold’ (package WGCNA (v1.72-1) function. A suitable soft threshold of 12 was selected. This matrix was then transformed into a TOM matrix to assess its connectivity within the network. The TOM matrix was used to construct a clustering dendrogram using average linkage hierarchical clustering. To ensure appropriate modules, the minimal gene module size was set to 10, and similar modules were merged if their threshold exceeded 0.25.

### 4.7. Statistical Analyses

Statistical analyses were conducted using Microsoft Excel (v2021). Data were presented as mean ± SEM. A *p*-value of less than 0.01 and an absolute log-fold change (|logFC|) greater than 2 were considered as indicators of significant differences.

## 5. Conclusions

Our comprehensive analysis of DEGs and DE lncRNAs in blood cell lineages sheds light on the unique expression profiles and regulatory networks within each cell type. Incorporating these studies into the context of our analysis, we emphasize the significance of deciphering gene expression patterns in distinct blood cell types and their related disorders. The convergence of findings from various investigations underscores the dynamic and intricate regulatory networks that govern hematopoiesis and highlights potential avenues for therapeutic interventions. The amalgamation of diverse studies enhances our understanding of the intricate mechanisms governing hematopoiesis and provides a foundation for future advancements in this field. Studying the roles and interactions of these common genes can provide valuable insights into the underlying mechanisms that govern the formation and behavior of different blood cell lineages. Understanding how these genes contribute to the specialized functions and characteristics of each cell type can deepen our knowledge of blood cell biology. Moreover, investigating the regulatory mechanisms of these shared genes may uncover potential therapeutic targets for various hematological disorders. Identifying specific genes that are crucial for blood cell development and function could offer new avenues for developing targeted therapies to treat blood-related diseases and conditions. Overall, this research has the potential to significantly advance our understanding of blood cell biology and pave the way for the development of novel and more effective treatments for hematological disorders, ultimately benefiting patients and improving their quality of life.

## Figures and Tables

**Figure 1 ijms-25-10073-f001:**
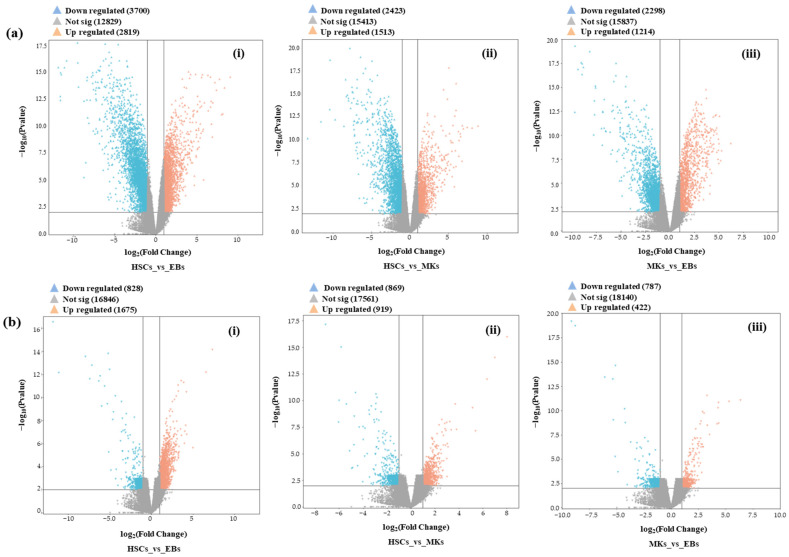
Detection of (**a**) DE mRNAs and (**b**) DE lncRNAs. DE mRNAs (**a**) and DE lncRNAs (**b**) are depicted in volcano plots. The volcano plots illustrate the dispersion of gene and lncRNA expression patterns across various blood cell lineages, encompassing HSCs, MKs, and EBs. These plots were generated based on the correlation between the negative logarithm of the *p*-value (on the Y-axis) and the logarithm of the fold change (on the X-axis). Each point on the plot represents an individual mRNA or lncRNA. Notably, mRNAs and lncRNAs highlighted in orange and blue indicate substantial fold changes (FC < −2 and >2) and statistical significance (adjusted *p* ≤ 0.01). Conversely, genes and lncRNAs represented in gray lack the required level of significance. (**i**) The plot showcases the expression profiles of mRNAs and lncRNAs between total HSCs and EBs. (**ii**) The plot displays the expression profiles of mRNAs and lncRNAs between total HSCs and MKs. (**iii**) Plot demonstrates the mRNAs and lncRNAs expression profiles between total MKs and EBs.

**Figure 2 ijms-25-10073-f002:**
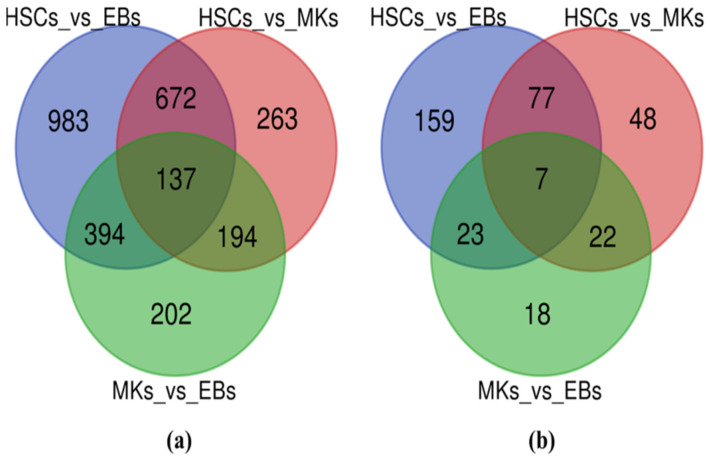
Identifying (**a**) DE mRNAs and (**b**) DE lncRNAs in HSCs, EBs, and MKs. Venn diagrams depicting the DE analysis of mRNAs and lncRNAs in HSCs, EBs, and MKs. (**a**) Venn diagram illustrating the distribution of DE mRNAs in HSCs, EBs, and MKs. Genes exhibiting an FC with an absolute value greater than 2 (|FC| > 2) and a *p*-value of ≤0.01 are highlighted in orange and blue, representing significant changes. (**b**) Venn diagram displaying the distribution of DE lncRNAs in HSCs, EBs, and MKs. Similar to panel (Figure 1a), lncRNAs with a |FC| > 2 and a *p*-value of ≤0.01 are shown in orange and blue, signifying significant alterations. The overlap of 137 DE mRNAs and 7 DE lncRNAs across the three datasets is depicted, indicating shared differential expression patterns among these cell lineages.

**Figure 3 ijms-25-10073-f003:**
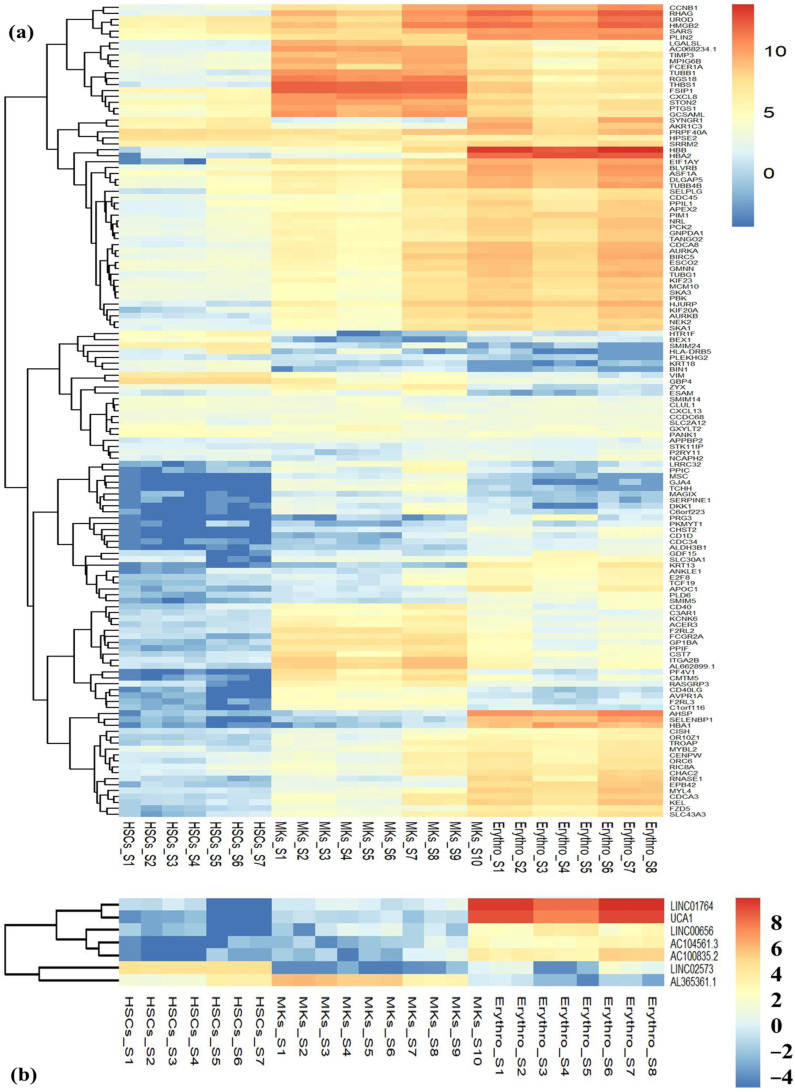
Hierarchical clustering of (**a**) DE mRNAs and (**b**) DE lncRNAs using their expression patterns. The presented heatmap illustrates the hierarchical clustering of shared (**a**) DE mRNAs and (**b**) DE lncRNAs, based on their expression profiles within three blood cell types (HSCs, EBs, and MKs). Across all three blood cell types, a total of 137 DE mRNAs and 7 DE lncRNAs were identified as differentially expressed (FC < −2 and >2, adjusted *p*-value ≤ 0.01), forming distinctive clusters of upregulated and downregulated genes. The Y-axis denotes the DEGs, while the X-axis represents sample IDs. The color gradient indicates the relative expression levels, where blue signifies low expression (downregulation), yellow represents moderate expression, and orange signifies high expression (upregulation). Each row corresponds to an mRNA or lncRNA, and each column corresponds to a specific blood cell type sample.

**Figure 4 ijms-25-10073-f004:**
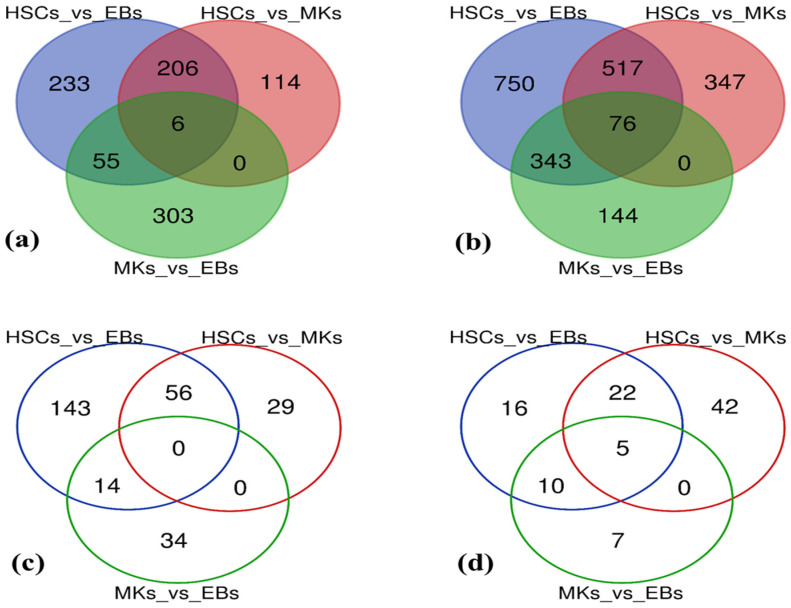
Distribution patterns of significantly upregulated and downregulated mRNAs and lncRNAs. The Venn diagram analyzed shared and distinct DE mRNAs and DE lncRNAs among HSCs, EBs, and MKs: (**a**) upregulated mRNAs, (**b**) downregulated mRNAs, (**c**) upregulated lncRNAs, and (**d**) downregulated lncRNAs. Overlapping circles represent common DE mRNAs and DE lncRNAs.

**Figure 5 ijms-25-10073-f005:**
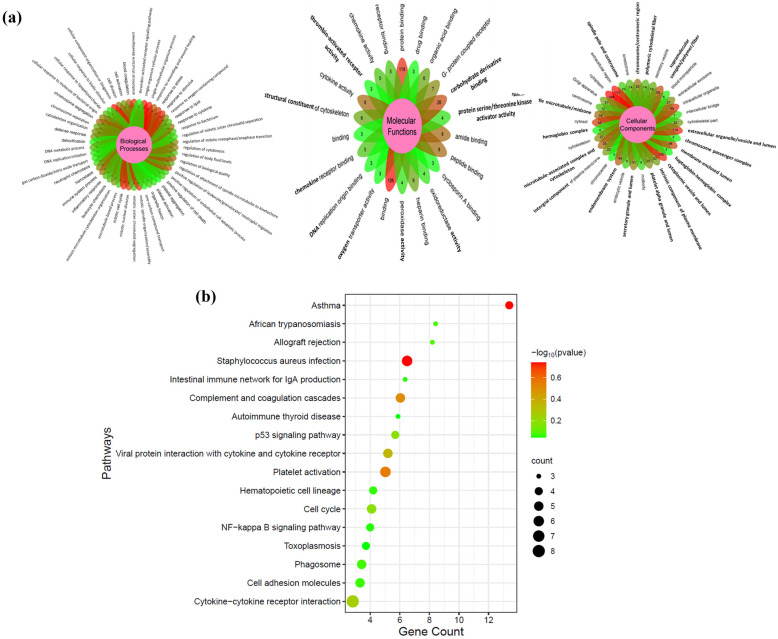
Enhancing understanding of DEGs in blood cell lineages: (**a**) GO (Appendix A). The varying shades of red and green colors signify different *p*-values (lower to higher) and (**b**) KEGG analysis. (**a**) Functional annotation was conducted using the bioinformatics tool DAVID. Flower plots reveal GO terms were categorized into molecular function, cellular component, and biological process. The most significantly enriched GO terms (FDR ≤ 0.05) in each branch are displayed. (**b**) The multi-group bubble plot represents top enriched KEGG pathways for DEGs in blood cell lineages (Appendix A). The varying shades of red and green in node colors signify different *p*-values (lower to higher), while the varying sizes of the nodes indicate varying numbers of genes. The most significantly altered pathways (FDR ≤ 0.05) include those associated with asthma, *Staphylococcus aureus* infection, platelet activation, and coagulation cascades.

**Figure 6 ijms-25-10073-f006:**
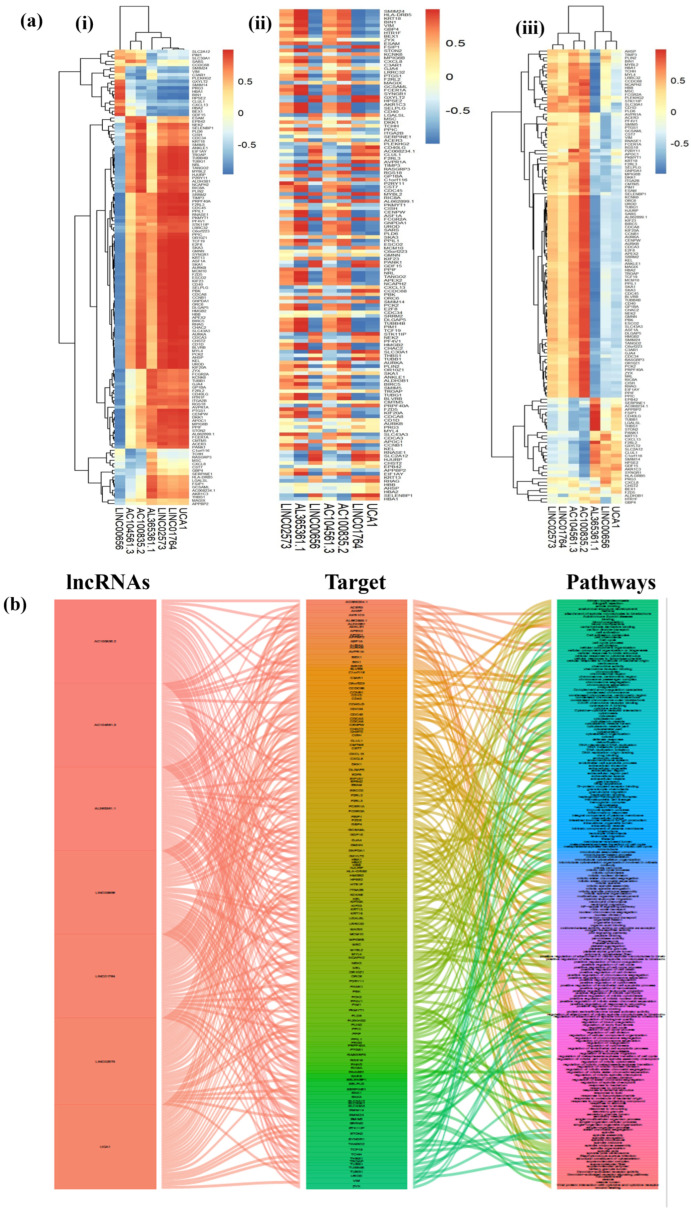
Correlation heatmaps between lncRNAs and their target genes and targeted pathways in blood cell lineages. (**a**) Cluster heatmaps of DE mRNA and DE lncRNA expression data. Hierarchical clustering utilized PCCs of log2-transformed FPKM expression values. (**i**) Correlation between lncRNAs and DE mRNAs in EBs, (**ii**) Correlation between lncRNAs and DE mRNAs in HSCs, and (**iii**) Correlation between lncRNAs and DE mRNAs in MKs. Colors on the scale represent correlation strength (BLUE: low correlation; RED: high correlation). (**b**) An alluvial plot reveals the compound-target-pathway network, highlighting common lncRNAs with significant correlations to their respective mRNAs or target genes. The plot consists of three columns: the left column represents lncRNAs, the middle column signifies significant interacting genes and the right column symbolizes pathways. Connections between them are represented by edges, with wider edges indicating a higher number of pathways-linked systems.

**Figure 7 ijms-25-10073-f007:**
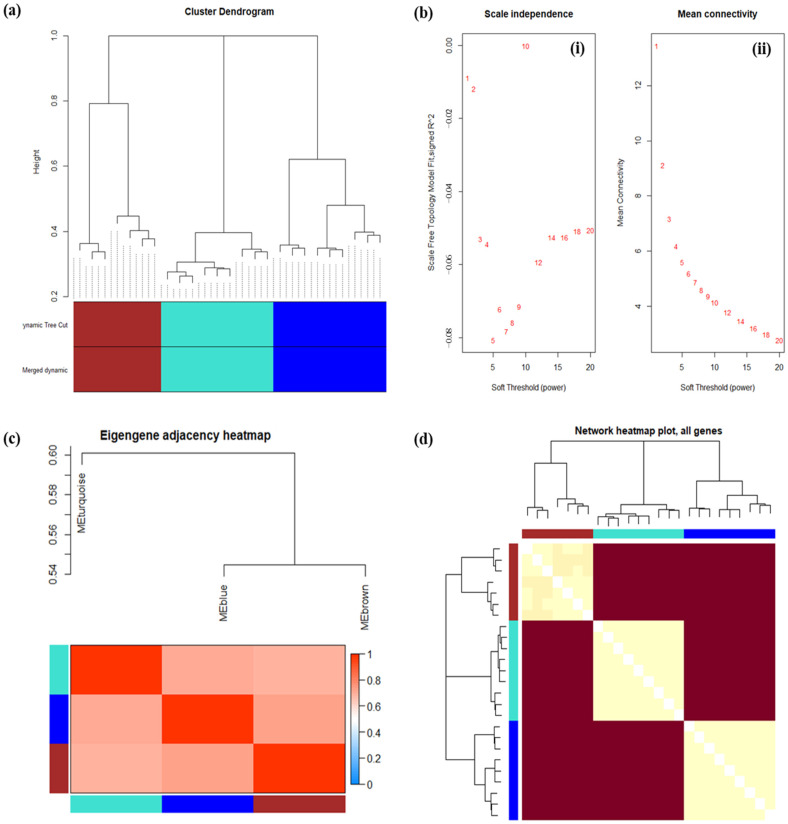
Multi-lineage WGCNA module analysis. (**a**) A cluster dendrogram is presented to illustrate the relationships among samples from the three cell lineages. The X-axis represents the height of the dendrogram, while the Y-axis displays the outcomes of two distinct clustering algorithms: dynamic tree cut and merged dynamic. Different colors indicate the clusters generated by these algorithms. Sample clustering was executed to identify outliers, and all samples were found within clusters, meeting the predefined cut-off thresholds. (**b**) Analysis of scale independence and mean connectivity was conducted to determine the appropriate soft-threshold power in WGCNA. The soft-thresholding power analysis aimed to achieve a scale-free fit index for network topology. The scale-free topology index and the mean connectivity for power values ranging from 1 to 20 are depicted in panels (**i**) and (**ii**), respectively. (**c**) The Eigen gene adjacency heatmap presents the relationships among various gene co-expression modules within blood cell lineages, revealing the connections between all the modules. (**d**) A heatmap showcasing the topological overlap within the gene network helps identify groups of correlated modules.

## Data Availability

The data are contained within the article and Appendix A.

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
