# Peer review of "Deciphering Transcriptomic Variations in Hematopoietic Lineages: HSCs, EBs, and MKs"

_ijms, 2024, doi:10.3390/ijms251810073_

Round 1
Reviewer 1 Report (New Reviewer)
Comments and Suggestions for Authors
Dahariya and collaborators present a work of great computational analysis complexity, utilizing different tools to perform a comprehensive analysis based on a broad and diverse experimental database (BLUEPRINT). Overall, the work is well-explained regarding the paths taken with the chosen computational tools for the analyses. However, some points require better detailing to indicate the line of reasoning chosen by the authors, as well as to provide details on the functionality of the chosen tool, explaining how the results are analyzed and obtained, and indicating the tool's analysis principle. For instance, the functional enrichment analyses of DEGs (GO and KEGG). In this analysis, the principle and the objective of these analyses are not clearly stated in the text. Additionally, the authors identified and thoroughly described thousands of genes, showing intriguing results when compared across the three cell types, but then proceeded to work with only 473 genes for these functional analyses. Although it becomes clear throughout the text that the p-value was used as a cutoff for most of the genes, this is not clearly described in the text as it should be. Moreover, regarding this analysis, the description of the sentence beginning on line 301 and ending on 303 seems contradictory to the final statement in the legend of Figure 5. I believe these two statements should be discussed together in the same text. The legend also lacks information on the meaning of the colors in item a. Additionally, the two methods of counting genes in Figure 5b were confusing.
The correlation analysis between coding genes and lncRNAs was undoubtedly one of the most striking aspects of the study for me. Although this part of the work is, in my view, the most intriguing due to the clear and functionally coherent behavior profile of lncRNAs, which is very homogeneous across the different DE genes plotted in MK and EB cells, this pattern was not observed in the original HSC cells (Figure 6a). However, this was not discussed. The discussion also falls short when mentioning that there are few lncRNAs described in the literature related to hematopoietic cells, without comparing these findings to the results obtained by the authors. I believe this study could greatly contribute to understanding the functional relationships between coding genes and lncRNAs. It seems that the authors encountered the challenge of performing meta-analyses with a large volume of information, making it difficult to provide clear direction. The biological processes identified are varied and related to a wide range of pathways in the cell's metabolic processes, making differential comparisons between cell types challenging due to the lack of specificity.
In general, data relevant to the discussion of hematopoietic cells were selected, but other coherent data, such as the large number of biological processes related to inflammatory issues mentioned early in the analysis, were left out without discussion or deemed unimportant. Poorly discussed data can undermine the accuracy and specificity of the work. Although the authors spend a significant portion of the discussion justifying potential shortcomings in the study due to the broad and diverse nature of the analysis and data collection, more relevant topics related to the study's objectives were neglected.
The final cluster analyses are speculative; I don't see an issue with that, as they could serve as suggestions for analyses that certainly require experimental validation, which should have been mentioned. More specific analyses of the results obtained would lead to a less evasive conclusion. In line 346, I believe the word "major" is not correctly applied. Furthermore, hematopoietic cells have a hierarchical formation process, starting with HSCs and branching into two main pathways: one myeloid and the other lymphoid. This hierarchy was not discussed in the paper, making it difficult to understand the purpose of the comparative objectives.
The interactions within this hierarchical process among the identified DE genes were not explored in the study. This comparative work, involving three cell types—two from the myeloid branch (MKs and EBs) and one original (HSCs)—did not clearly explain why the authors specifically chose these cell types and, for example, excluded granulocytes, which are also part of the myeloid branch.
The amount of work presented here is substantial, and I commend the authors for that, but they need to clarify some points of reasoning and better focus the discussion to make the study less evasive.
Author Response
Provided as separate word document addressing the reviewer's concerns.

Reviewer 2 Report (New Reviewer)
Comments and Suggestions for Authors
In this manuscript, the Authors have used available datasets to investigate long-non coding RNA (lcnRNA) and mRNA expression in specific hematopoietic compartments, including stem cells (HSCs), erythroblasts (EBs), and megakariocytes (MKs). The argument is of interest in hematopoietic biology field, however some improvements are required.
1. The introduction must be shortened by removing unnecessary examples, as well as results must be more concisely presented.
2. Tables should be moved as supplementary, as there are no informative data.
3. There are some highlighted parts.
4. Quality of some figures must be improved (e.g., Figures 3, 5, and 6).
5. Figure 7 is very difficult to follow, even by reading the legend or the text. Please improve or change visualization type.
Comments on the Quality of English Language
Minor English errors.
Author Response
Provided as a separate document.

Round 2
Reviewer 2 Report (New Reviewer)
Comments and Suggestions for Authors
The Authors have addressed all comments.
This manuscript is a resubmission of an earlier submission. The following is a list of the peer review reports and author responses from that submission.
Round 1
Reviewer 1 Report
Comments and Suggestions for Authors
The paper utilizes publicly available genomics data to investigate the correlation between mRNA and lncRNA in differentiating blood cells of HSCs, EBs, and MKs. The previous comments have been addressed, but two remaining questions need clarification:
1. Explain the discrepancy in line 20-22 “This scrutiny revealed a total of 27,982 messenger RNAs (mRNAs), with a notable revelation that 14,233 of them were previously unidentified or novel, highlighting a substantial reservoir of genetic information yet to be explored. ” and line 140-142 “By examining the expression profiles of 3 distinct hematopoietic lineages - we identified a total of 13,749 mRNAs and 3,092 lncRNAs. ”
2. How can the public benefit from the findings of the paper? How can these finding be used by other researchers in the field.
Reviewer 2 Report
Comments and Suggestions for Authors
This manuscript presents a comprehensive exploration of the genetic traits of HSCs, EBs, and MKs using transcriptomic analyses, which is commendable for its ambition and scope. The discovery of numerous novel mRNAs and the differential expression of lncRNAs in cell differentiation offer valuable insights into hematopoietic cell biology.
However, there are several areas in the manuscript that require attention to enhance its scientific rigor and clarity:
Section 2.2 (Differential Gene Definition Methods)
The manuscript does not specify whether p-values have undergone Benjamini-Hochberg correction in context, which is critical for validating the statistical significance of differential gene expression observed.
Section 2.3 (Application of DEGs to GO and KEGG Analysis)
It is unclear which DEGs were applied to the GO and KEGG analyses and from which cell type comparisons these DEGs were derived. This lack of clarity hampers the understanding of the gene sets' relevance to specific cellular contexts.
The KEGG analysis results with a maximum -log10 p-value of 0.7 do not meet the usual significance threshold of 1.3 (FDR < 0.05), suggesting that the pathway enrichment analysis lacks meaningful significance.
Section 2.4 (WGCNA Analysis)
The manuscript fails to clarify the association between WGCNA color blocks and the DEGs. Additional details on the DEGs included in each WGCNA module would aid in understanding their biological implications.
In addition, the use of WGCNA based solely on DEGs is questioned, as it typically requires analysis across the entire genes. The scale-free topology index values reported as negative, indicating the network does not meet the fundamental conditions for a scale-free network (typically, a scale-free topology index of 0.8 or higher is desirable).
Section 4.2 and 4.3 (Analysis Methods of DEGs)
The use of both Limma and DESeq2 for differential expression analysis is noted, but the manuscript does not clarify which results were obtained using which package. A consistent approach using one method would enhance the comparability of the results.
Section 4.7
The manuscript mentions comparing two groups using Student's t-test or Mann-Whitney U test, yet there is no comparative data between two groups presented. This statement is confusing and needs clarification regarding its application.
Other Issues
Several figures have text that is too small to be legible (e.g., Figures 3a, 5a, 6), and some legends are cropped or incomplete (e.g., Figure 3b), which could be improved for better clarity and understanding.
The significance of different colors (red and green) in Figure 5a is not explained, necessitating a detailed legend to interpret these visual cues correctly.
Comments on the Quality of English LanguageNA
Round 2
Reviewer 2 Report
Comments and Suggestions for Authors
The author response lacks any meaningful revisions addressing the review comments, including concerns regarding the non-significant enrichment in KEGG analysis and issues with font size. Additionally, the WGCNA figures do not show satisfactory explanations or corrections.
Comments on the Quality of English LanguageNA